# Light Interception and the Growth of Pastures under Ideal and Stressful Growing Conditions on the Allegheny Plateau

**DOI:** 10.3390/plants9060734

**Published:** 2020-06-11

**Authors:** Edward B. Rayburn, Thomas C. Griggs

**Affiliations:** 1Extension Service, West Virginia University, Morgantown, WV 26506-6108, USA; 2Division of Plant and Soil Sciences, West Virginia University, Morgantown, WV 26506-6108, USA; Thomas.Griggs@mail.wvu.edu

**Keywords:** cool-season grasses, legumes, forage mass, forage growth rate, growth curves, root density, light interception, pasture

## Abstract

Pasture-based livestock production is impacted by management and weather. In pastures, there is conflict between leaf retention for plant growth and leaf harvest for animal nutrition. Defoliated pastures with low light interception (LI) may have a low forage growth rate (FGR), while excessive growth shades leaves, reducing FGR and resulting in an S-shaped regrowth curve. To optimize production, it is best to keep FGR linear. Three studies were conducted to evaluate the impact of management and weather on FGR. Replicated pastures were used to measure FGR when grazed from 25 to 10 cm and allowed to regrow. The impact of alternative defoliation timings and intensities on FGR were studied using clipped treatments at three recovery intervals and two stubble heights. Variability in FGR was studied using a field validated plant growth model. Of the 24 growth periods studied, two displayed exponential, 12 linear and 10 linear-plateau growth. There was no effect of FM on growth curve form. In May and June, LI increased with canopy height, up to 0.93. Stubble height and days of growth impacted FGR with an interaction. There was no treatment impact on root density. Weather caused variation in FGR. A low FGR risk occurs at high elevations; greater risk occurs east of the plateau.

## 1. Introduction

The predominant agriculture in the hilly lands of the Allegheny Plateau of eastern North America is pasture-based ruminant livestock production. Perennial mixed-species pastures are used to produce beef and dairy cattle, sheep and goats on land not suitable for row crop production. Due to the latitude and elevation of the Allegheny Plateau, the climate has a moderate temperature and good rainfall, and is best suited for the production of cool-season grasses, legumes and forbs, such as tall fescue *Lolium arundinaceum* (Schreb.) S.J. Darbyshire (previously known as *Festuca arundinacea* Schreb.), orchardgrass (*Dactylis glomerata* L.), timothy (*Phleum pratense* L.), quackgrass (*Elytrigia repens* L.), bluegrass (*Poa*), bentgrass (*Agrostis*), white clover (*Trifolium repens* L.), red clover (*Trifolium pratense* L.), plantain (*Plantago lanceolata* L.), dandelion (*Taraxacum officinale* F.H. Wigg.) and chicory (*Cichorium intybus* L.).

The Allegheny Plateau is a region that varies in elevation and climate. On the west side of the plateau, rainfall increases as elevation increases from west to east. Rainfall decreases rapidly once past the east side of the Allegheny front (the escarpment on the east side of the plateau). This front also marks the western edge of the Appalachian Ridges and Valleys. Mean daily temperatures decrease with increasing elevation, then increase as elevation decreases on the east side of the Allegheny Front.

Light interception (LI) by plant leaves is used in photosynthesis to provide energy for plant maintenance, to grow new leaves and roots, and to produce carbohydrates stored within cells and translocated to energy sinks. In pastures, the plant leaves needed for LI and photosynthesis are the plant part harvested by livestock for food. This conflict between leaf retention for plant growth and leaf harvest for animal production requires a balance of pasture growth and defoliation to optimize plant and animal production.

Net photosynthesis is measured as net carbon exchange (NCE) per unit leaf area, and is a function of the photosynthetically active radiation (PAR, 400–700 nm), leaf age, leaf nutrient status, light environment during leaf development, current light environment, adequate plant available water (PAW) in the rooting zone to ensure carbon dioxide diffusion through leaf stomates and optimal air temperature. In a pasture canopy, NCE is a function of the sward’s leaf area index (LAI), LI of ambient PAR and NCE efficiency. Typically, LI is measured as a function of LAI. In forage crops, forage mass (FM) and canopy height are highly correlated to LAI, and can be used as alternatives to LAI for estimating LI [1,2,3,4]. Pasture FM is measured as kg of plant dry matter (DM) per hectare.

For the cool-season forages typical of the Allegheny Plateau, high temperatures can reduce photosynthetic efficiency through photorespiration. Even though the Allegheny Plateau receives abundant rainfall, periods of dry weather occur, where plants experience short-term droughts that limit carbon dioxide diffusion into leaves for photosynthesis and limit cell expansion for growth. Plant growth is the result of positive NCE moderated by available soil moisture. For ideal plant growth, all environmental conditions need to be optimal. These include air temperature, PAW, soil fertility and total PAR availability due to daylength, solar elevation and cloud cover. Variations in the weather impact FGR, causing uncertainty about the number of animals that can be supported on a farm in a given year or season of the year.

When pastures are grazed, the leaf area consumed by livestock is no longer available for LI and photosynthesis. When a sward is excessively defoliated, leaf area is suboptimal for LI and photosynthesis and growth may be adversely impacted. With the excessive accumulation of FM, the upper horizons of dense swards intercept most of the incident light (LI > 0.95), shading lower leaves, which may result in leaf death and reduced growth. This is the basis for the concept of an S-shaped growth curve occurring after pasture defoliation. When plant growth is controlled by LI and photosynthesis, theoretically, the growth of new FM takes the form of a sigmoidal curve (third order regression) over time (T), where residual FM (RFM) is the FM at the end of the previous grazing event:FM = RFM + b T + c T^2^ − d T^3^(1)

This curve divides into three sections (Figure 1):early exponential growth (FM = +c T^2^, section A to B),mid linear growth (FM = +b T, section B to C) andlate plateau growth (FM = −d T^3^, section C to D).

Another way to visualize pasture growth is the daily change in FM. Forage growth rate (FGR, kg DM ha^−1^ day^−1^) is the first derivative of FM at a given point, where time is in days:FGR Day^−1^ = b + 2 c Day − 3 d Day^2^(2)

For example:Forage mass at a point in time: FM = 1000 + 1 Day + 7 Day^2^ − 0.11 Day^3^Forage growth rate at a point in time: FGR Day^−1^ = 1 + 14 Day − 0.33 Day^2^

During the growing season, FGR needs to be in balance with livestock feed demand. When FGR exceeds daily feed demand, there will be excess forage available for harvest or to be managed for late season grazing; a practice referred to as ‘stockpiling’. A primary pasture management goal is to optimize animal and plant performance through proper timing and intensity of grazing to control production and utilization of forage. This may be accomplished best by keeping pasture in the linear growth phase. Another pasture management principle is budgeting forage growth relative to livestock feed demand over the year. However, FGR is not constant, and varies due to environmental conditions which are not always optimal for growth.

Three questions arise:Do current grazing timing and intensity recommendations maintain pastures in a linear growth phase, or do they cause a depression in early or late regrowth?What impact does alternative defoliation timing and intensity have on LI, plant growth and plant health, as measured by rooting activity?What impact does weather variation across the Appalachian Plateau have on pasture growth over the growing season?

## 2. Results

### 2.1. Pasture Growth Rate after Recommended Defoliation

Grazing management prior to the test growth period resulted in the initial growth being a little taller than planned, but the final height was on goal (Table 1). Litter treatments had greater CHt, RHt and FM at the start and end of the growth period than the no-litter treatments. There was no treatment effect on FM grown during the growth period. Litter treatments grew faster than the no-litter treatments and produced more forage per day.

### 2.2. CHt, LI and Growth Curve

Multiple regression analysis found that, of the 24 growth periods, two (8%) displayed exponential, 12 (50%) linear and 10 (42%) linear-plateau growth (Table 2). The difference in frequency of occurrence of linear vs. linear-plateau growth due to fertility level was significant (*P* = 0.02). No growth periods displayed the entire range of sigmoidal growth. The FM at the beginning and end of growth were not significantly different by growth curve forms (Table 2) and there was no effect of either FM on growth being linear (*P* = 0.87) or plateau (*P* = 0.62).

Pastures receiving poultry litter displayed a high frequency of linear growth while pastures receiving no-litter displayed a high frequency of linear-plateau growth (Table 2). This may be due to litter providing a high level of available nitrogen (N), which maintained active growth, while pastures not receiving litter had lower levels of available N, resulting in a plateau of growth as N was tied up in the canopy.

Pastures receiving no-litter had a slower FGR (Table 1). Average pasture FGR was 36 kg ha^−1^ d^−1^ for litter treated pastures and 24 kg ha^−1^ d^−1^ for no-litter treated pastures (*P* = 0.02). Pastures not treated with litter required more days to reach the goal grazing height. This resulted in later growth occurring after spring moisture was used, resulting in a growth plateau, possibly due to soil moisture stress.

Poultry litter is a good source of N fertilizer, and increased FGR as expected [5]. That the addition of litter would increase linear growth and reduce linear-plateau growth was not anticipated. The implemented grazing management, defoliation at a sward CHt of 14.2 to 16.8 cm (23.4 to 27.4 cm RHt), resulted in a similar at-harvest PAR fractional LI of 0.90 to 0.93 for the no-litter and litter treatments, respectively. The forage crude protein content of pastures receiving litter was higher than for those not receiving litter (204 vs. 167 g kg^−1^, respectively, ±5), as would be expected due to the greater N available in these pastures. The lower crude protein content in the pastures not receiving litter would be reflected in lower chlorophyll content and lower photosynthetic and growth rates, especially toward the end of the growth cycle, resulting in the plateau of growth. This is supported by the fact that dead leaf material in these pastures increased as the season progressed (0.08 ± 0.04 in May to 0.51 ± 0.09 Oct) and was higher in pastures not receiving litter than in pastures receiving litter (0.52 ± 0.03 vs. 0.20 ± 0.03) across the growing season. Nitrogen is a mobile nutrient within plants. When a pasture is deficient in N, N will be translocated from mature lower leaves to young upper leaves, resulting in greater senescence and death in the lower canopy. Low N status in the pastures not receiving litter resulted in the linear-plateau growth curves being dominant in these pastures.

The Gompertz growth model was not useful for evaluating pasture growth curve form. The Gompertz model is misleading in that it always calculates a slow growth phase, which may not be present. The slow growth phase calculated by the Gompertz model is dependent on the range of dates in which data are collected. When data collection is delayed, the calculated slow growth stage is delayed, and the estimate of FM associated with slow growth increases (Figure 2).

As canopy height increased, LI increased to a CHt of 17 cm, plateauing at 0.93 of ambient PAR (Figure 3). Across the growing season, LI exceeded 0.90 only in the first and second growth periods, and for other periods ranged from 0.30 to 0.70 (Figure 4).

Both LI and FM increased with sward height (Figure 5A); LI reached a peak at a CHt of 18 cm (30 cm RHt), and FM reached a peak at a CHt of 23 cm (38 cm RHt). The efficiency of LI per unit FM was low at a CHt below 8 cm (13 cm RHt), peaked at 10 cm (17 cm RHt), then decreased as CHt increased to 25 cm (42 cm RHt) (Figure 5B).

### 2.3. Impact of Defoliation Timing and Intensity on Plant Growth and Rooting Density

Under simulated grazing, stubble height (*P* = 0.04) and days of growth (*P* = 0.03) affected the total FM available for grazing (Figure 6 and Table 3), and there was a stubble height by days regrowth interaction (*P* = 0.05). The 9-day continuous stocking produced a higher yield at 7 cm than the 13 cm stubble height, while the other regrowth intervals produced equivalent yields at the 7 cm and 13 cm stubble heights.

Canopy LI was impacted by days of growth (*P* < 0.001) but not by stubble height (*P* = 0.16), and there was no interaction (*P* = 0.63, Table 3). The portion of pasture growth produced after June 21 was impacted by days of growth but not stubble height. The 35-day, 9-day and 44-day growth treatments produced 41, 48 and 35 percent of their annual yield after June 21.

Defoliation interval and intensity had no impact on root density at the 0–7.5 and 7.5–15 cm depths, but there was a sampling date effect and date by depth interaction (Figure 7). Root density was impacted by depth (*P* < 0.001), with the 0–7.5 cm depth having a root density of 4.4 g kg^−1^ soil vs. the 7.5–15 cm depth having a root density of 0.7 g kg^−1^ soil (SE = 0.3).

### 2.4. Growth Variability and Risk in Stressful Environments

Based on the stochastic pasture growth model, climate and weather cause considerable variation in pasture FGR within the region (Figure 8). Calculated mean annual yields were 9.0 metric tons ha^−1 1^ at Morgantown and 9.4 metric tons ha^−1^ at Terra Alta. The variability of FGR at Morgantown was higher than at Terra Alta, based on the greater SD about the mean. It can be inferred that a similar average stocking rate can be supported at these locations, but more adjustment will be needed from year to year at Morgantown than at Terra Alta. The calculated mean annual yield at Moorefield was 7.6 metric tons ha^−1^. This indicates that a lower average stocking rate can be supported at Moorefield than at Morgantown or Terra Alta. Pasture growth at Moorefield had a higher SD about the mean than occurred at Morgantown and Terra Alta. This indicates that livestock producers near Moorefield must also be more skilled in managing stocking rates and grazing pressures from year to year, as weather patterns differ. Historically, and to a lesser extent today, livestock producers in the Moorefield area move livestock to higher plateau areas similar to Terra Alta for the summer to manage this climatic effect.

Other than site-specific soil fertility, the primary limiting factors for pasture growth in the Allegheny plateau are rainfall, temperature and daylength. Although the active growth period for Terra Alta is the shortest, because of cool temperatures in spring and fall, the total yield estimates for Terra Alta are highest because of more consistent amounts and distribution of rainfall, and little to no temperature stress on cool-season forages.

## 3. Discussion

The pasture growth available for grazing is a function of the available PAR, LI, NCE, abiotic stress and change in stored energy reserves [6,7,8].
G_Leaf_ = G_Eff_ (Photo_Eff_ * PAR_Int_ − Night_Res_ − G_root_ − ΔSER)(3)

G_Leaf_Growth of leavesG_Eff_Growth efficiency {ƒ(plant nutrition, temperature and moisture stress)}Photo_Eff_Photosynthetic efficiency {ƒ(plant nutrition, temperature and moisture stress)}PAR_Int_PAR interception {ƒ(LAI or sward RHt, solar elevation, and daylength)}Night_Res_Night respiration {ƒ(night temperature)}G_Root_Growth of roots {ƒ(net photosynthesis, temperature and moisture stress)}ΔSERChange in stored energy reserves

Growth efficiency is dependent on plant nutrition [5], especially N, P and K, as well as temperature and moisture stress [6,7,8]. Photosynthetic efficiency is dependent on plant nutrition, leaf age [9,10,11], temperature and moisture stress. High temperatures increase photorespiration in cool-season forages, reducing NCE [7].

As a pasture canopy grows in height, it produces more FM and LAI, increasing LI [1,12,13,14,15]. PAR interception is also dependent on solar elevation (SE). At solar noon, SE varies with the month due to the declination of the earth. Most of the month’s effect on LI is removed by viewing LI as a function of solar beam path length (PL) through the canopy: PL = CHt/sin (SE)(4)As PL increases (low SE), LI increases:LI = 0.54 Ln (PL) − 0.07 Month   R^2^ = 0.99, SD_reg_ = 0.08, N = 209(5)
where Month = 1 for the month of May

Month = 0 for all other months

Canopy PL accounts for all months except May. May has a lower LI, since grass tillers become reproductive, with elongating internodes elevating leaves in the canopy.

Through the day, the SE ranges from zero at sunrise to its maximum at solar noon. The maximum solar elevation (SE_max_) is a function of latitude and declination [16]. In northern latitudes
SE_max_ = 90 − latitude + declination(6)

The intensity of the direct beam PAR on a horizontal surface (PAR_HS_) increases with SE: PAR_HS_ = PAR * sin (SE)(7)

Integrating total daily LI from direct beam radiation on the 21st day of the month, accounting for SE across the day and day length, the effect of sward height on total daily LI can be calculated (Figure 9). On June 21, the canopy needs a 15 cm CHt (25 cm RHt) to maximize LI. The lowest LI occurs in October, at 52% of that in June, requiring a 10 cm CHt (17 cm RHt) to maximize LI.

In the Allegheny Plateau of West Virginia, sunlight provides 560 to 1410 PAR units or more at solar noon 50% of the time (Table 4). During the growing season, the PAR is less than 290 units 10% of the time, and greater than 1700 units 10% of the time. This range in intensity is due to cloudy days, earth declination and day length. The canopy LI of combined diffuse and direct beam PAR is greater than direct beam PAR by 5% to 8% when PAR is below 1580 µmol of photons m^−2^ s^−2^ on cloudy days.

In July and August, high daytime temperatures can be stressful to cool-season plants. When this occurs, it benefits pastures to maintain a high canopy height to keep light transmission low, shading the soil to keep it cool. When the weather is cool (a sunrise temperature of less than 16 °C) this is not a problem, and pastures may be grazed to a lower residual height to manage for increased tiller density and legume content to fix nitrogen and improve forage quality.

There is a need for more research on the impact of cloudy, rainy weather on forage quality and animal performance. Beef calf weaning weights across the region have decreased by 23 kg head^−1^ in rainy, cloudy summers (1 year out of 20), apparently due to decreased forage quality. Predictions of climate change suggest that the weather on the Allegheny Plateau will become warmer, cloudier and wetter, which may make reduced animal performance more frequent.

When pasture is rotationally stocked at high grazing pressure, removing much of the FM and LAI, the canopy will be dependent on carbohydrate (CHO) reserves to power the initial plant growth [14,17,18,19,20,21]. Root growth is impacted by plant partitioning energy usage. With defoliation, new leaf growth has priority for energy reserves over root growth.

### Balancing Leaf Growth and Harvest

The timing and intensity of defoliation is the key management practice needed to optimize pasture growth and harvest. As time progresses, leaves grow using CHO reserves, LI increases and leaves age, reducing NCE and forage quality. Initially, CHO reserves decrease, then start to increase as LAI and NCE increase. Timing and intensity also impact tiller dynamics. As the degree of defoliation increases, tiller size decreases and tiller density increases. A greater leaf harvest results in less LAI and more dependence on CHO reserves for regrowth.

The optimal timing and intensity of defoliation is dependent on the forage species in the canopy. Perennial ryegrass (*Lolium perenne* L.), harvested about every 4 weeks to residual heights from 20 to 100 mm, produced the most regrowth when harvested at a residual height of 56 mm. The total water soluble CHO content per tiller was reduced with increasing defoliation, with peak herbage yield and water soluble carbohydrates occurring at a 60 mm residual height [17]. Under rotational stocking, grazing from 27 to 7 cm produced 23% more harvested forage than 20 to 5 cm, due to more orchardgrass, quackgrass and tall growing legumes in the pasture [22].

In a 2-year comparison of continuous stocking, rotational stocking (2 days of grazing monthly to 15 cm residual sward height; six grazing cycles) and haymaking (two cuts) on mixed cool-season grass-legume pasture, Oates et al. [23] observed higher potential season-total utilizable forage under rotational stocking than under continuous stocking and haymaking (11,500, 7250, and 7750 kg DM ha^−1^, respectively). Root masses from 15 cm deep in-growth cores sampled each October were higher for haymaking than for rotational and continuous stocking (2000, 1250, and 950 g m^−2^, respectively) in one year but not different in the second year (2600, 2500, and 1700 g m^−2^ for haymaking, rotational and continuous stocking, respectively). While these treatment responses differ from ours, Hart et al. [24] found in a 3-year comparison of orchardgrass and tall fescue, clipped weekly to 5, 10 or 20 cm RSH or every 28 days to 5 cm RSH, that total-season forage mass was higher for monthly clipping at 5 cm than for weekly clipping at any height. Within weekly clipping treatments, season-total forage production decreased with increasing RSH, which is consistent with our results for simulated continuous stocking at 7 vs. 13 cm RSH (Treatments C and D, Table 3 and Figure 6).

Other comparisons of infrequent hay-stage defoliation to more-frequent defoliation under clipping or rotational stocking treatments were made by Brink et al. [25,26]. In a 2-year trial [25], plots of meadow fescue (*Schedonorus pratensis* (Huds.) P. Beauv.), tall fescue and orchardgrass were clipped infrequently (every 40–65 days; three cuts) or frequently (whenever sward height reached ~25 cm; six cuts) and at 5 or 10 cm RSH. Across treatments and years, annual forage production was greater for infrequent than for frequent harvesting, and for 5 cm than for 10 cm RSH. Our results show little to no effect of RSH, with the exception of Treatment C relative to D, but otherwise show greater forage production for infrequent than for frequent clipping (Table 3). In a second 2-year trial [26], meadow fescue, orchardgrass, quackgrass and reed canarygrass (*Phalaris arundinacea* L.) paddocks were rotationally stocked at vegetative (32 cm height) or mature (48 cm height) stages. Vegetative paddocks were grazed to an RSH of 16, 8 or 2 cm, and mature plots were grazed to an RSH of 24, 12 or 4 cm. In the first year, the annual utilization of vegetative paddocks by cattle to 2 or 8 cm RSH did not differ, but was greater for one or both of these shorter RSH than for 16 cm RSH. In a second year with wetter and warmer conditions, annual utilization again did not differ for the two shorter RSH, except for quackgrass, and there were fewer differences in utilization for the shorter than for the 16cm RSH. In mature paddocks in both years, annual utilization decreased with increasing RSH in both years, with a few exceptions. In most cases, utilization was lower for 12 than for 4 cm RSH. Our data are consistent with these findings for simulated continuous stocking at 7 vs. 13 cm RSH, but otherwise show little impact of RSH on productivity.

In mixed species pastures, species diversity is beneficial for N fixation by the legume component, and for stabilizing forage production by deep rooted forbs and legumes in dry weather [27,28,29]. Stocking rate can override the influence of species diversity when overgrazing eliminates more productive species.

High LI results in good growth and forage yield. However, high LI results in the thinning of tiller density and reduced tiller ground cover. Good forage management is a balance between forage yield, forage digestibility, tiller density and plant diversity. Long rest intervals and high LI increase annual forage production, whereas short rest intervals and lower LI increase stand health when measured as tiller ground cover [25].

Clover leaves start growth at ground level, while orchardgrass and tall fescue leaves come out of tillers about 5 cm RHt above ground level. Legume leaves increase photosynthesis in proportion to light as PAR increases from 100 to 1000 units (light compensation point vs. light saturation point) [2]. Providing adequate light transmission into a pasture stimulates legume growth [1], which is the primary source of N in the pasture ecosystem.

The late fall grazing of mixed-species pastures increased tiller number, reduced tiller size and doubled the number of white clover growing points compared to early closed paddocks, but reduced the total nonstructural carbohydrates [29].

This study was conducted on the Allegheny Plateau of eastern North America. However, the results apply to other temperate humid areas of the world with similar climatic conditions where pasture-based livestock production is a primary agricultural practice.

## 4. Materials and Methods

The hypotheses posed in these three questions were addressed though three experiments.

### 4.1. Pasture Growth Rate after Recommended Defoliation

To measure the effect of current grazing recommendations on pasture growth, two replications of four rotationally grazed pastures (eight pastures) were measured for sward height once a week during the growing season for three years. The experiment site was on the West Virginia University, Reedsville Experiment Station, Reedsville, WV (39°1 N, −79°82 W), on the western side of the Eastern Allegheny Plateau and Mountains (Table 5), on a Gilpin silt loam soil (Typic Hapludult Ultisols). The forage species in these swards included tall fescue, orchardgrass, timothy, quackgrass, bluegrass, red clover, white clover and diverse forbs. Grazing management was based on allowing pasture sward growth to reach a goal ruler height (RHt) of 20 to 25 cm, then having cattle graze the pasture to a goal residual ruler height of 8 to 10 cm. Within the replications, two pastures received poultry litter at 4500 or 9000 kg^−1^ ha^−1^ yr^−1^, and two pastures received no poultry litter. All pastures received lime applications when soil pH fell below 6.0.

To evaluate pasture FGR response to LI under near ideal growing conditions, only the second growth period each season was used. In periods other than the second growth period, environmental conditions other than LI determined FGR. In the first growth period, FGR was controlled by soil and air temperatures being suboptimal. Growth later in the season was controlled by variable rainfall, and occasionally by high air temperatures.

Second growth events started between May 6 and June 20, and continued for 19 to 90 days depending on the year, weather and fertility level. In the last year of the study, a major drought in the latter part of the growth period impacted FGR due to low soil moisture.

Sward height was measured as canopy compressed height (CHt) using a standard resting plate meter [30,31]. This meter is an acrylic plastic plate measuring 45.7 by 45.7 cm square, 5.6 mm thick, weighing 1332 g. Sward CHt was measured by holding the plate meter, using strings that support the plate, and a 0.91 m pasture stick in one hand. The tip of the end of the stick was placed on the soil surface, with the stick held vertical. The plate was lowered gently onto the sward until the sward was holding up the entire weight of the plate. The height at the top of the plate (CHt) was measured to the nearest half cm.

Canopy CHt was measured at 30 points, selected at random within each pasture, along a walked zig-zag sampling transect. Producers often measure pasture height using a ruler, giving RHt. The resting plate meter CHt is a less subjective measure than RHt. These two methods can be cross-calibrated to translate height measurements between these methods [32]. To cross-calibrate RHt and CHt, RHt was measured using the plate meter, with the plate lowered onto the pasture canopy until three out of four of its quadrants touch a plant leaf, with the strings holding up the entire weight of the plate. The height at the top of the plate was read as RHt. The plate was then lowered onto the canopy until the canopy held up the entire weight of the plate, and CHt was measured as previously described. Producers measuring RHt were asked to define RHt as the height of the tallest leaf within 10 cm of the pasture stick to approximate the cross-calibration. The cross-calibrations between RHt and CHt are:CHt = 0.63 RHt − 0.4 cm  approximated by CHt = 0.6 RHt(8)
RHt = 1.48 CHt + 1.8 cm  approximated by RHt = 1.6 CHt(9)

When calibrating CHt vs. FM, forage samples were clipped at ground level. For each sampling event, FM (kg DM ha^−1^) was calculated from the average CHt (cm) using calibration equations developed for these pastures [32,33]:FM = 308 CHt − 6.47 CHt^2^(10)

Canopy LI at average CHt was calculated from calibration equations developed for these pastures for the May through June sampling period. Each LI vs. CHt data point was the average of four above- and four below-canopy ground-level PAR readings, and two CHt readings [1]:When CHt <= 17 cm:  LI = −0.357 + 0.389 CHt − 0.0293 CHt^2^(11)
When CHt > 17 cm:  LI = 0.93(12)

Peak LI was 0.93 of ambient light above the canopy, and occurred in a 17 cm CHt sward.

The pasture growth curve was evaluated using sequential multiple regressions. For each growth event, the dependent variable FM was regressed against days regrowth using a third order model (Equation (1)). Regression coefficients were tested for significance (*P* = 0.05), and variables with non-significant coefficients were removed. This was repeated until only variables with significant regression coefficients remained. Pasture growth curves were also evaluated using the theoretical non-linear Gompertz growth model [34]. The effect of fertility level, and FM at the beginning and end of the growth period on growth curve form was evaluated with a Chi-square test. Statistical analysis was conducted using NCSS 11 [35].

### 4.2. Impact of Defoliation Timing and Intensity on Plant Growth and Rooting Density

This experiment used mechanical clipping to simulate pasture defoliation intervals and intensities or severities under continuous and rotational stocking. We hypothesized that moderate defoliation (every 35 days) at low intensity (13 cm residual height) would produce greater forage mass than more (every 9 days) or less (every 45 days) frequent or intense (7 cm residual height) defoliation.

This experiment was conducted at the West Virginia University Reedsville Research Experiment Station (Table 5) [36] on a Latham silt loam (fine, mixed, semiactive, mesic aquic Hapludults). Plots measuring 7 by 3 m were delineated in existing mixed perennial cool-season grass-legume and forb pastures in June 2016. In early July 2016, six simulated grazing treatments described below were initiated at three defoliation frequencies and two levels of severity across three blocks in a randomized complete block design. Continuous-frequency plots were defoliated every 7–11 days, moderate-frequency (pasture-stage) rotational plots were defoliated every 28–35 days and infrequent hay-stage (mature) plots were defoliated every 42–45 days. At each frequency, plots were clipped to residual stubble heights of 7 cm (more severe) and 13 cm (less severe). Plots were clipped with a walk-behind sickle-bar mower with a 0.91-m serrated sickle bar (Jari USA, St. Peter, MN, USA). Clipped forage remained on each plot.

To determine defoliation management impacts on pasture LI, a line quantum sensor (ACCUPAR LP-80 Ceptometer, Decagon Devices Inc. Pullman, WA) equipped with an external quantum sensor was used to measure the intensity of PAR above the canopy and within the canopy at 2 cm above soil surface level. Below- and above-canopy PAR readings were used to calculate the fraction of incident PAR intercepted by the canopy. Measurements of PAR LI were made at three random locations within a plot within 1.5 h of solar noon in most cases. Readings were taken at the approximate mid-point or end of each treatment growth period.

To determine the impact of defoliation strategies on forage yield, canopy RHt was measured with a meter stick at 10–12 random locations within each plot before and after each clipping. A similar set of 14 readings of sward CHt was taken with an Ellinbank-type rising plate meter with a square aluminum plate measuring 0.1 m^2^ in area [37].

The descriptions of treatments simulating rotational, continuous and hay-stage stocking are:Rotational, low stubble: clip to 7 cm after regrowth to 25–30 cm or 35 days.Rotational, high stubble: clip to 13 cm after regrowth to 25–30 cm or 35 days.Continuous, low stubble: clip to 7 cm at 7- to 11-day intervals or regrowth to 10–11 cm.Continuous, high stubble: clip to 13 cm at 7- to 11-day intervals or regrowth to 15–16 cm.Hay-stage, low stubble: clip to 7 cm after growth to mature hay stage or 42–45 days.Hay-stage, high stubble: clip to 13 cm after growth to mature hay stage or 42–45 days.

Soil cores for the determination of root density were obtained in duplicate from each plot with a hammer-driven sampling tube on June 19 and October 12. Each 4.4 cm-diameter by 15 cm-long core was divided into upper (0–7.5 cm) and lower (7.5–15 cm) horizons, and air dried. Samples were weighed, then roots were separated from soil by washing over a screen with 850 µm aperture size. Samples were oven-dried at 60 °C, weighed, and root density was expressed as g root kg^−1^ soil. Data are from the second year of treatments on each plot.

### 4.3. Variability of Pasture Growth across the Allegheny Plateau in West Virginia

A plant growth model based on local weather history and a Penman-type evapotranspiration model was used as a stochastic model to evaluate the impact of variations in weather on FGR in three climatic zones of the Allegheny Plateau and Northern Appalachian Ridges and Valleys (Table 5). This model uses inputs of latitude, day of the year, daily rainfall, and daily maximum and minimum temperature [8]. Potential evapotranspiration (PET) is calculated as a function of potential solar radiation for the day of the year [16], modified by estimated cloud cover based on the daily temperature range (maximum minus minimum temperature). Calculated PAW in the rooting zone is the sum of the previous day’s PAW plus daily rainfall, minus PET. The effect of PAW and air temperature on the growth of cool-season forages was used to modify the expected FGR. This model was field tested using within-pasture weather and clipped forage production over 16-site years. The model’s predicted growth compared to clipped forage growth had an R^2^ of 0.81 and a residual SD of 964 kg ha^−1^ across the growing season.

National weather data from local weather stations were used to calculate the probability of a day having precipitation (Table 6), and when a day had precipitation the probability of how much rain fell on days with rain (Table 7). These probabilities were used to make the model a stochastic growth model using daily time steps starting on January 1, with PAW at field capacity. The model was run for 12 annual cycles, and the mean and SD of daily FGR were calculated.

The effect of climate and weather on FGR was modelled for three sites located from west to east along a line near 39° N Lat. Sites differed in elevation above sea level and location within the greater Allegheny Plateau region (Table 5). Morgantown is west of the high elevation plateau, whereas Terra Alta is near the top of the high elevation plateau and Moorefield is east of the Allegheny Front, in the rain shadow of the high elevation plateau. The differences in climate between these sites can be seen in the mean July temperature (Table 5), monthly rainfall, and the probability of it raining two days in a row and being dry two days in a row (Table 6). The growing season average gamma distribution of rainfall amount on days with rain for the three sites is presented in Table 7.

## 5. Conclusions

In the grazing study on pasture regrowth curves, beginning and ending FM did not influence growth curve form. Fertility management did affect the frequency of the growth curve form, with pastures receiving poultry litter displaying more linear growth, and pastures not receiving litter displaying more plateau growth.

The Gompertz growth model was not useful for measuring forage growth dynamics. This model always calculates a slow growth phase, even when one does not exist, causing a systematic error in predictions. The Gompertz model uses the lowest observed FM to estimate a slow growth phase. Under all conditions, the multiple regression approach is flexible, and returns a measure of significant coefficients for growth across the range of growth observed.

In the simulated grazing study, regrowth interval and harvest intensity impacted the forage growth available for grazing and LI, but not plant root density. The lack of root response to defoliation treatments may have been due to the inherent variability of root cores from mixed-species plots, and/or the insufficient duration of treatments over the years to detect treatment responses.

In the stochastic model comparison, climate (elevation within the plateau) and weather impacted pasture growth distribution and variability. The canopy height needed to optimize LI over the day is lower than that needed at solar noon, since the PL followed by the solar beam increases as SE decreases. This has the most significance in fall growth due to low SE and when preparing the sward for winter and the following spring’s growth.

## Figures and Tables

**Figure 1 plants-09-00734-f001:**
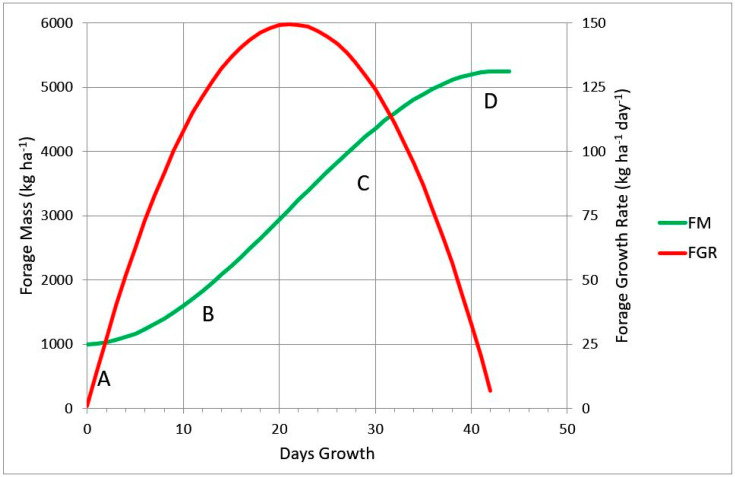
Pasture growth based on a third order (cubic) regression of forage mass (FM, kg DM ha^−1^) vs. time (days), starting with the residual FM (RFM) remaining at the end of the previous grazing event (FM = RFM + b Day + c Day^2^ − d Day^3^) and the associated forage growth rate (FGR, DM ha^−1^ day^−1^) based on the first derivative of the cubic regression (FGR = b + 2 c Day − 3 d Day^2^), where RFM is 1000, b is 1, c is 7 and d is 0.11.

**Figure 2 plants-09-00734-f002:**
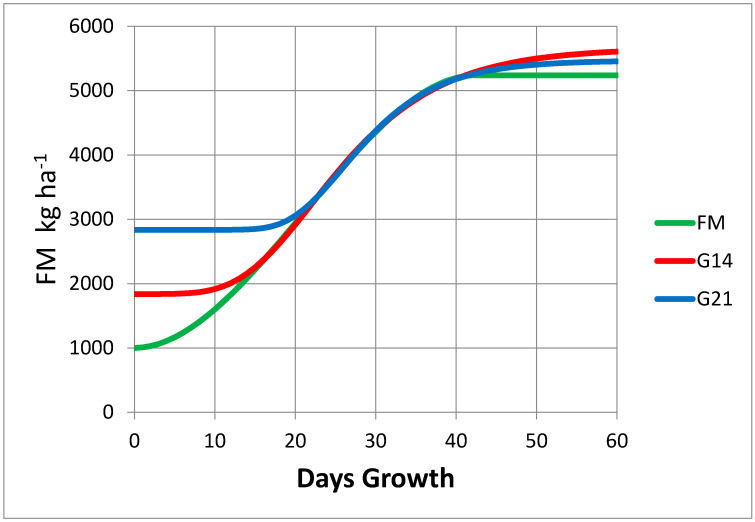
The Gompertz non-linear growth model always calculates an initial slow growth phase. This becomes a systematic error when there is no slow growth phase. The forage mass (FM) line represents the ‘true’ growth rate for a theoretical pasture. The Gompertz model, evaluated over days 14 to 43 (G14) and days 21 to 43 (G21), predicted zero growth at an FM of 1900 and 2900 kg DM ha^−1^, respectively, while the ‘true’ growth had a low growth rate at an FM of 1000 kg DM ha^−1^.

**Figure 3 plants-09-00734-f003:**
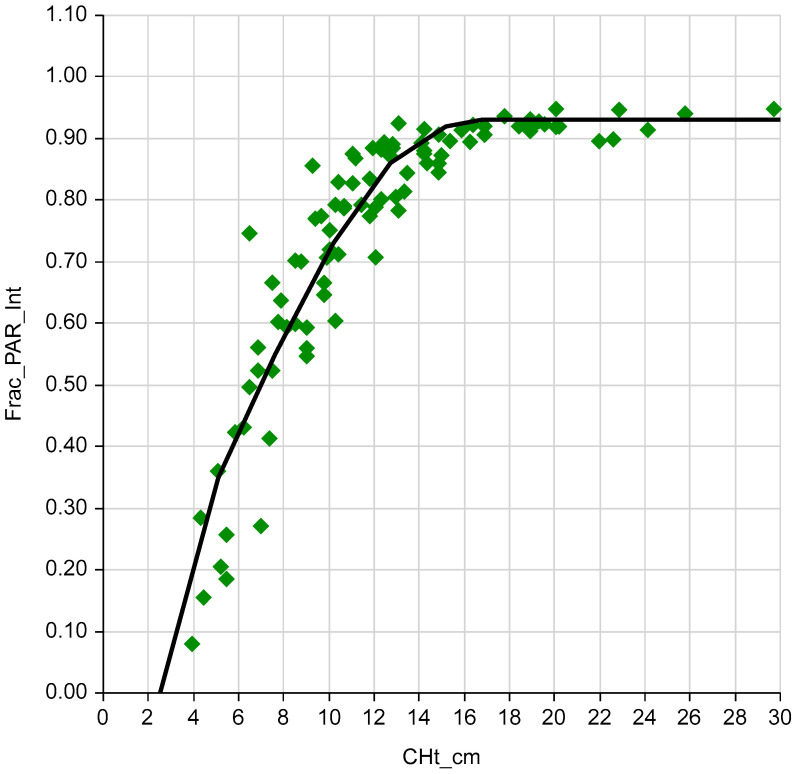
Interception of photosynthetically active radiation (LI, fraction of ambient PAR above canopy) in May and June, relative to pasture compressed height (CHt, cm).

**Figure 4 plants-09-00734-f004:**
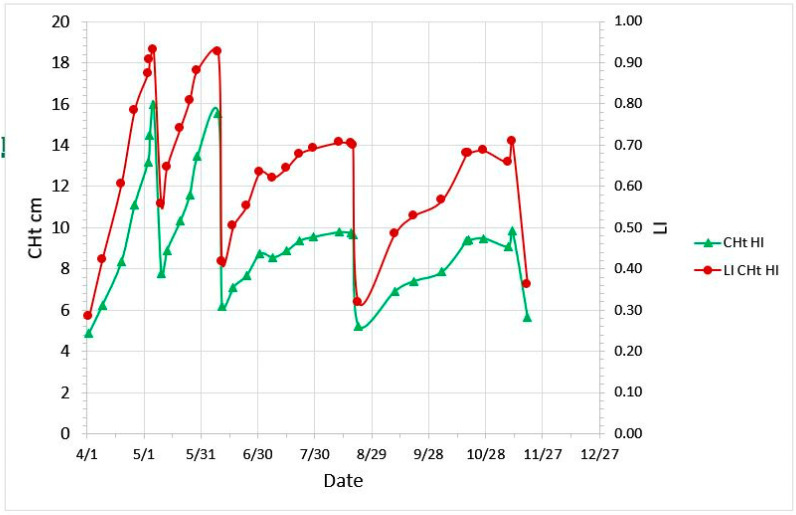
Pasture plate meter compressed height (CHt, cm) and light interception (LI, fraction of ambient PAR) across the grazing season, showing growth and defoliation of the canopy.

**Figure 5 plants-09-00734-f005:**
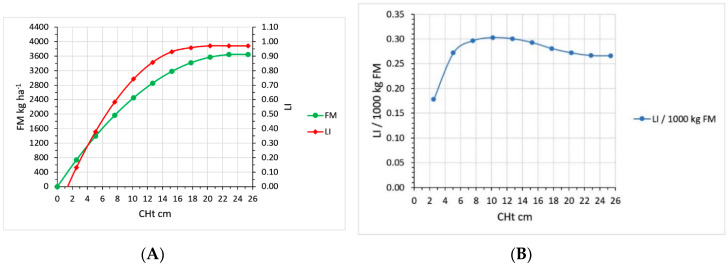
(**A**) Forage mass (FM, kg ha^−1^) and light interception (LI, fraction amdient PAR) increase as pasture canopy plate meter compressed height (CHt, cm) increases. (**B**) At low CHt, LI efficiency per FM was low, increasing as CHt increased to 8 cm, then decreased slightly as CHt increased to 25 cm.

**Figure 6 plants-09-00734-f006:**
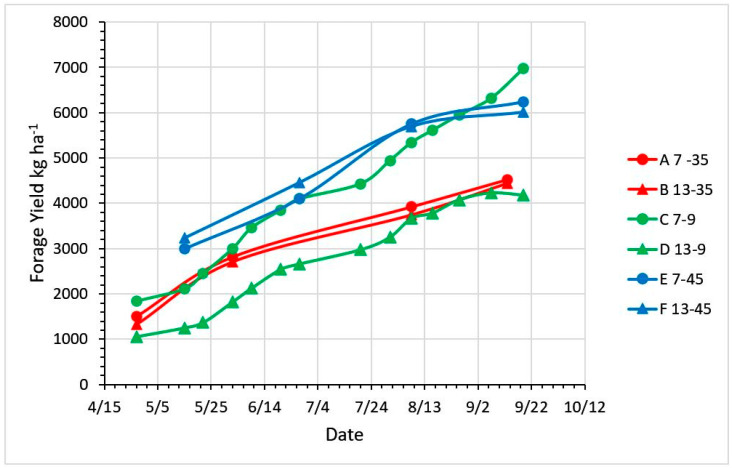
Growth of pasture swards managed under two residual stubble heights (7 and 13 cm) and three cutting intervals, simulating moderate rotational stocking (35 days), continuous stocking (9 days) and infrequent rotational stocking (hay-stage, 45 days) strategies.

**Figure 7 plants-09-00734-f007:**
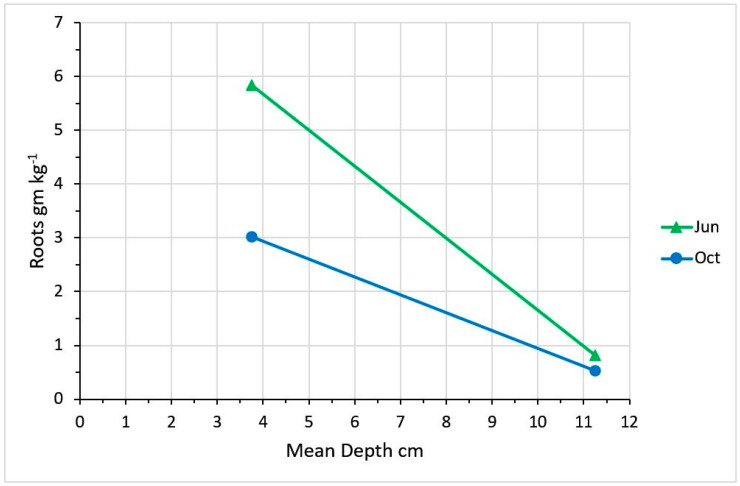
Root mass at mean soil depths for samples taken at 0 to 7.5 and 7.5 to 15 cm depths. Each data point represents the mean of 18 samples.

**Figure 8 plants-09-00734-f008:**
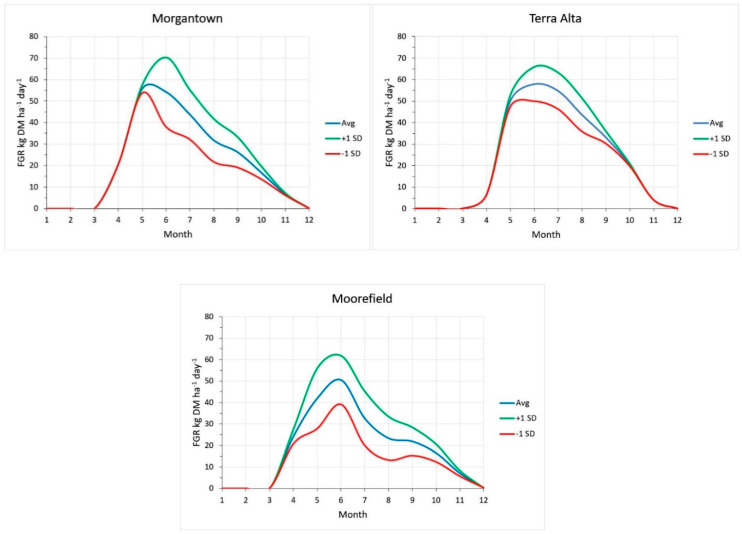
Average and variability (±1 SD) of pasture growth at three sites in the Allegheny Plateau and Mountains, and Northern Appalachian Ridges and Valleys.

**Figure 9 plants-09-00734-f009:**
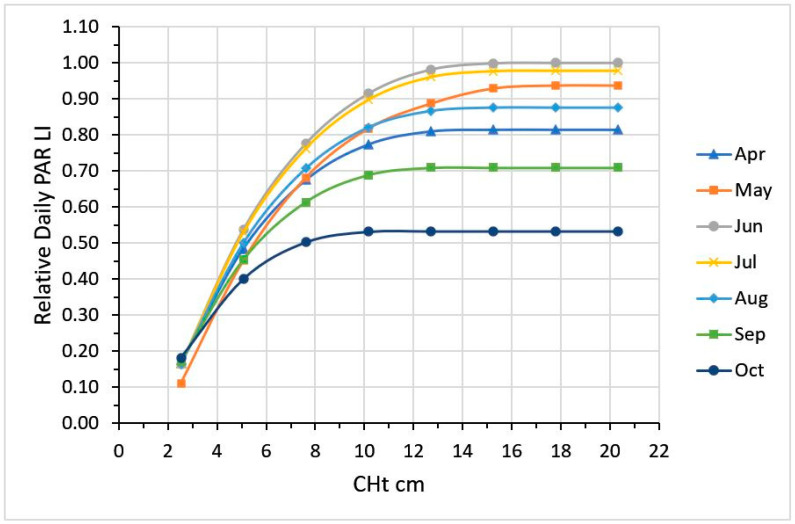
Relative total daily PAR intercepted by pasture canopies differing in plate meter compressed height (CHt, cm) due to solar elevation over the day and daylength at 39° N Lat.

**Table 1 plants-09-00734-t001:** Canopy characteristics and litter treatments’ effect (mean ± SE) on pastures’ initial and final plate meter compressed height (CHt_min_ and CHt_max_), initial and final ruler height (RHt_min_ and RHt_max_), days in the growth period (days growth), initial and final forage mass (FM_min_ and FM_max_), forage mass grown (FM grown) and forage growth rate per day (FGR day^−1^).

Measure	Units	Litter	No-litter	*P*
CHt_min_	cm	8.6 ± 0.5	7.4 ± 0.5	0.05
CHt_max_	cm	16.8 ± 0.8	14.2 ± 0.8	0.05
RHt_min_	cm	14.2 ± 0.8	11.9 ± 0.8	0.05
RHt_max_	cm	27.4 ± 1.3	23.4 ± 1.3	0.05
Days growth	Days	35 ± 6	56 ± 6	0.02
FM_min_	kg DM ha^−1^	2162 ± 90	1901 ± 90	0.05
FM_max_	kg DM ha^−1^	3272 ± 96	2991 ± 96	0.05
FM grown	kg DM ha^−1^	1111 ± 52	1091 ± 52	0.78
FGR day^−1^	kg DM ha^−1^ day^−1^	36 ± 3	24 ± 3	0.02

**Table 2 plants-09-00734-t002:** Frequency of occurrence of growth curve form for pastures growing in May and June by fertility management (no poultry litter vs. litter at 4.5 or 9 metric tons ha^−1^ yr^−1^) vs. all pastures (N = 24). Mean (±SE) residual forage mass (RFM) at the end of the previous grazing and growth achieved at the end of regrowth (FM) (T in days from start of growth).

Fertility Level	Growth Curve Form		
	Exponential FM = RFM + b T^2^	Linear FM = RFM + b T	Linear-Plateau FM = RFM + b T − c T^2^
Frequency of occurrence of growth curve form			
Litter	1 (4%)	9 (38%)	2 (8%)
No-litter	1 (4%)	3 (12%)	8 (33%)
Total	2 (8%)	12 (50%)	10 (42%)
Forage mass at start and end of growth by growth curve form			
RFM (*P* = 0.34)	2239 ± 172	2086 ± 72	1920 ± 132
FM (*P* = 0.14)	3286 ± 109	3246 ± 54	2958 ± 149

**Table 3 plants-09-00734-t003:** Simulated pasture harvest treatments’ impact on the total forage growth available for grazing, with light interception (LI) as a fraction of ambient PAR and average pasture forage growth rate (FGR, kg DM ha^−1^ day^−1^).

Treatment	Stubble Height cm	Days Growth	Forage Available for Grazing kg DM ha^−1^	Different from	PAR LI	Different from	FGR	Different from
A	7	35	4526	C	0.91	C	21	C, D, E, F
B	13	35	4445	C	0.94	C	21	C, D, E, F
C	7	9	6975	A, B, D	0.77	A, B, D, E, F	35	A, B, D, E, F
D	13	9	4176	C, E	0.86	C, F	24	A, B, C
E	7	44	6242	D	0.94	C	27	A, B, C
F	13	44	6008		0.96	C, D	23	A, B, C

**Table 4 plants-09-00734-t004:** Cumulative probability distribution of the photosynthetically active ration (PAR, μmol of photons m^−2^ s^−2^) at solar noon in West Virginia being at or below the indicated level (based on 1333 measurements over two years, rounded to the nearest 10 units).

Month	N	Percent of Days			Day Length Mid-Month, Hours
		25	50	75	
Apr	120	580	1360	1630	13.1
May	257	690	1350	1670	14.2
Jun	191	720	1410	1580	14.8
Jul	272	700	1310	1580	14.5
Aug	170	470	900	1460	13.5
Sep	183	570	1010	1200	12.2
Oct	140	130	560	1150	10.9

**Table 5 plants-09-00734-t005:** Climatic description (Mean ± SD), elevation and location for the two experiments and the three sites, evaluated stochastically for pasture growth.

Site	County	Ann. Rainfall mm	January Temp. C	July Temp. C	Elevation m	MLRA
Experimental site						
Reedsville	Preston	1255 ± 373	−3 ± 4	20 ± 1	537	127
Stochastic model sites						
Morgantown	Monongalia	1067 ± 178	−1 ± 3	23 ± 1	305	126
Terra Alta	Preston	1422 ± 203	−3 ± 3	20 ± 1	792	127
Moorefield	Hardy	838 ± 127	0 ± 3	23 ± 1	305	147
Major Land Resource Area (MLRA, USDA/NRCS, 2006) 126 Central Allegheny Plateau 127 Eastern Allegheny Plateau and Mountains 147 Northern Appalachian Ridges and Valleys						

**Table 6 plants-09-00734-t006:** Growing season rainfall and rainfall probabilities used in the stochastic pasture growth model: probability that there will be two days in a row with rain (rain today it will rain tomorrow, RTD RTM), probability that there will be a day with rain and a day without rain in a row (rain today it will not rain tomorrow or visa versa, RTD NRTM), probability that there will be two dry days in a row (no rain today it will not rain tomorrow, NRTD NRTM); and gamma probability distribution alpha and beta coefficients for the amount of rain occurring on days when it does rain.

Month	Avg Rainfall (mm)	RTD RTM	RTD NRTM	NRTD NRTM	Alpha	Beta
Morgantown						
Apr	90	0.25	0.40	0.35	0.73	0.35
May	99	0.24	0.38	0.38	0.69	0.41
Jun	99	0.20	0.37	0.43	0.70	0.46
Jul	104	0.17	0.41	0.42	0.73	0.47
Aug	102	0.15	0.38	0.47	0.64	0.56
Sep	84	0.15	0.35	0.50	0.78	0.43
Oct	65	0.15	0.34	0.51	0.58	0.42
Terra Alta						
Apr	119	0.32	0.37	0.31	0.80	0.37
May	127	0.28	0.38	0.34	0.80	0.43
Jun	132	0.25	0.39	0.36	0.76	0.50
Jul	152	0.22	0.41	0.37	0.68	0.65
Aug	116	0.21	0.38	0.41	0.58	0.62
Sep	98	0.18	0.37	0.45	0.72	0.46
Oct	85	0.18	0.34	0.48	0.72	0.42
Moorefield						
Apr	59	0.10	0.30	0.60	1.09	0.28
May	85	0.15	0.36	0.49	0.73	0.44
Jun	90	0.13	0.35	0.52	0.75	0.49
Jul	94	0.12	0.34	0.54	0.89	0.45
Aug	85	0.10	0.34	0.56	0.79	0.50
Sep	65	0.08	0.30	0.62	0.82	0.45
Oct	62	0.07	0.23	0.70	0.80	0.51

**Table 7 plants-09-00734-t007:** For the three locations used for the stochastic modeling of pasture growth, the probability of daily rainfall on days when it does rain from April through October, based on historic precipitation patterns, expressed as a gamma distribution of daily rainfall amount, using growing season average alpha and beta coefficients.

Probability	Morgantown	Terra Alta	Moorefield
	Rain Fall mm		
10	0.3	0.5	0.8
25	1.5	1.8	2.3
50	4.6	5.3	5.8
75	10.7	12.2	13.0
90	19.6	22.4	22.4

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
