# Peer review of "Light Interception and the Growth of Pastures under Ideal and Stressful Growing Conditions on the Allegheny Plateau"

_plants, 2020, doi:10.3390/plants9060734_

Round 1

Reviewer 1 Report

Rayburn and Griggs tested the impact of management and weather on the forage regrowth rates of pasture. The authors question whether current recommendations on grazing time and intensitiy influence the shape of the forage regrowth curves. Furthermore, they tested the impact of alternative defoliation timing and intensity on light interception, plant growth, and plant health as measured by rooting activity, as well as the impact of wheather variaton across the Appalachian Plateau do have on pasture growth over the season. In generell, the topic of forage regrowth rates is of interest to the readeship of Plants. However, I wonder what statement do the authors want to give to the reader not living in the Allegheny Plateau area? The authors found that pastures receiving poultry litter displayed a higher frequency of linear growth than without fertilizing, which seems not very surprising, or?

To me the first part of the Discussion rather belong to the Results part since it includes a table and a figure.

Figure 1: Legend is not clear to me.

Author Response

We have modified Figure 1 and the legend to Figure 1 to improve readability.

We are leaving Figure 9 and Table 4 in the discussion since they are part of the discussion and explain how the research applies to management and are not part of the conducted field research.

Point 1 Text added to discussion as suggested by Reviewer 3 to answer question of Review 1

This study was conducted on the Allegheny Plateau of eastern North America. However, the results apply to other areas of the world having similar climatic conditions where pasture-based livestock production is a primary agricultural practice.

Point 2 Text added to the (results ? / discussion ? section) in answer to Reviewer 1 questions and comments on the effect of poultry litter.

Poultry litter is a good source of nitrogen fertilizer and increased pasture growth rate as expected. That the addition of litter would increase Linear growth and reduce Linear-Plateau growth was not anticipated. The implemented grazing management, defoliation at a sward CHt of 14.2 to 16.8 cm (23.4 to 27.4 cm RHt) resulted in similar at harvest PAR fractional LI of 0.90 to 0.93 for the no-litter and litter treatments, respectively. Forage crude protein content of pastures receiving litter was higher than for those not receiving litter (0.0204 vs 0.0167 mg kg-1, respectively, ± 0.0005) as would be expected due to the higher available nitrogen in these pastures. The lower crude protein content in the pastures not receiving litter would be reflected in lower chlorophyll content and lower photosynthetic and growth rate, especially toward the end of the growth cycle, resulting in the plateau of growth. This is supported by the fact that dead leaf material in these pastures increased as the season progressed (0.08 ± 0.04 in May to 0.51± 0.09 Oct) and was higher in pastures not receiving litter than in pastures receiving litter (0.52± 0.03 vs 0.20 ± 0.03). Nitrogen is a mobile nutrient within plants. When a pasture is deficient in nitrogen, nitrogen will be translocated from mature lower leaves to young upper leaves, resulting in greater senescence and death in the lower canopy. The low nitrogen status in pastures not receiving litter resulted in the linear-plateau growth curves being dominant.

Other:

We added a reference that was missing in the reference list but cited.

  1. Earle, D.F.; A.A. McGowan. Evaluation and Calibration of an Automated Rising Plate Meter for Estimating Dry Matter Yield of Pasture. Aust. J. Exp. Agric. Anim. Husb. 1979, 19, 337-343, doi:10.1071/EA9790337.

Reviewer 2 Report

The manuscript entitled "Light Interception and Growth of Pastures Under 3 Ideal and Stressful Growing Conditions on the Allegheny Plateau" is a well written manuscript fall into the scope of the journal. 

Author Response

Thank you.

Reviewer 3 Report

First comment (but not to the Authors, rather to Editors): something went wrong with the title of the manuscript. It was announced in an email as „Light Interception and Growth of Pastures Under Ideal and Stressful Growing Conditions on the Allegheny Plateau” – and (as I suppose) it is the right title, however, in the file I have received for review it is entitled „Regulatory potential of bHLH-type transcription factors on the road to rubber biosynthesis in Hevea brasiliensis”, what is an obvious mistatke.

I consider the manuscript as interesting and well presented.

It describes the results obtained for one special region, Allegheny Plateau of eastern North America, but it can be representative for many similar regions of the world, where the main agriculture is pasture-based ruminant livestock production. Such studies are essential to keep this type of agriculture in balance with the environment, and to prevent progressive deterioration of pastures. Thus, the manuscript put understandable theoretical basis for the proper pasture management. These reasons could not be enough for the publication of this manuscript in a scientific journal concerning plants, however, it contains a lot of data directly connected to plant physiology and plant response to stress. Performed experiments have been well-designed and quite laborious, results are clearly presented and refered to theoretical mathematical models explained in the introduction.

Author Response

Thank you for your comments and suggestions.